# The Impact of Glucose-Based or Lipid-Based Total Parenteral Nutrition on the Free Fatty Acids Profile in Critically Ill Patients

**DOI:** 10.3390/nu12051373

**Published:** 2020-05-11

**Authors:** Pavel Skorepa, Ondrej Sobotka, Jan Vanek, Alena Ticha, Joao Fortunato, Jan Manak, Vladimir Blaha, Jan M. Horacek, Lubos Sobotka

**Affiliations:** 1Department of Military Internal Medicine and Military Hygiene, Faculty of Military Health Sciences, University of Defence in Brno, Trebesska 1575, 50001 Hradec Kralove, Czech Republic; pavel.skorepa@unob.cz (P.S.); vanek.conf@gmail.com (J.V.); jan.horacek@unob.cz (J.M.H.); 23rd Department of Internal Medicine—Metabolic Care and Gerontology, University Hospital and Faculty of Medicine in Hradec Kralove, Charles University in Prague, Sokolska 581, 50005 Hradec Kralove, Czech Republic; SobotkaO@lfhk.cuni.cz (O.S.); Joao.fortunato@lfhk.cuni.cz (J.F.); manak@lfhk.cuni.cz (J.M.); blaha@lfhk.cuni.cz (V.B.); 3Department of Physiology, Faculty of Medicine in Hradec Kralove, Charles University in Prague, Simkova 870, 50003 Hradec Kralove, Czech Republic; 4Department of Clinical Biochemistry and Diagnostics, University Hospital and Faculty of Medicine in Hradec Kralove, Charles University in Prague, Sokolska 581, 50005 Hradec Kralove, Czech Republic; alenaticha@fnhk.cz; 54th Department of Internal Medicine—Hematology, University Hospital Hradec Kralove, Sokolska 581, 50005 Hradec Kralove, Czech Republic

**Keywords:** critical illness, parenteral nutrition, nonesterified fatty acids, essential fatty acids deficiency, adiponectin, alpha-Tocopherol, insulin resistance

## Abstract

Introduction: Our study aim was to assess how the macronutrient intake during total parenteral nutrition (TPN) modulates plasma total free fatty acids (FFAs) levels and individual fatty acids in critically ill patients. Method: Adult patients aged 18–80, admitted to the intensive care unit (ICU), who were indicated for TPN, with an expected duration of more than three days, were included in the study. Isoenergetic and isonitrogenous TPN solutions were given with a major non-protein energy source, which was glucose (group G) or glucose and lipid emulsions (Smof lipid; group L). Blood samples were collected on days 0, 1, 3, 6, 9, 14, and 28. Results: A significant decrease (*p* < 0.001) in total FFAs occurred in both groups with a bigger decrease in group G (*p* < 0.001) from day 0 (0.41 ± 0.19 mmol∙L^−1^) to day 28 (0.10 ± 0.07 mmol∙L^−1^). Increased palmitooleic acid and decreased linoleic and docosahexaenoic acids, with a trend of increased mead acid to arachidonic acid ratio, on day 28 were observed in group G in comparison with group L. Group G had an insignificant increase in leptin with no differences in the concentrations of vitamin E, triacylglycerides, and plasminogen activator inhibitor-1. Conclusion: Decreased plasma FFA in critically ill patients who receive TPN may result from increased insulin sensitivity with a better effect in group G, owing to higher insulin and glucose dosing and no lipid emulsions. It is advisable to include a lipid emulsion at the latest from three weeks of TPN to prevent essential fatty acid deficiency.

## 1. Introduction

Nutritional support is an integral part of the complex treatment of critically ill patients, especially in those who are under long-term care at the intensive care unit (ICU). Total parenteral nutrition (TPN) is essential for the provision of the patient’s energy requirements together with macro and micronutrients in critically ill patients when enteral feeding is contraindicated or not tolerated [1]. Despite the efforts of many experts, the exact doses of glucose and lipids (or their ratios) during the provision of nutritional support have not been completely elucidated and are still being rigorously discussed [2,3]. Recent guidelines also admit that even lipid-free parenteral nutrition may be given during the first week of critical illness. In case there is a suspicion of an essential fatty acid deficiency (EFAD), lipids should be given at a maximum of 100 g per week [4]. Nevertheless, there is no precise information about the dosage or timing of essential fatty acids supplementation [1]. In spite of the fact that some of the present nutritional recommendations are based on prospective randomized studies and their meta-analyses, the pathophysiological context of such recommendations could, on occasion, be put into question [5].

The hydrolysis of triglycerides derived from both the diet and adipose tissue determine the concentration and composition of plasma free fatty acids (FFAs). Elevated levels of FFAs are frequent in obese subjects and especially in those who are obese and insulin-resistant or in those who are already type 2 diabetics [6]. Recently, Arabi et al., in their study, described a positive relationship between the elevated levels of FFAs and blood glucose with increased insulin requirements in critically ill patients with diabetes and in older patients, thus suggesting a connection between elevated FFA levels and metabolic syndrome [7]. It was previously shown that therapeutic procedures that indirectly influence the levels of FFAs could be important in some clinical states. The increase of serum FFA levels due to the administration of lipid emulsion in healthy volunteers led to an elevated inflammatory response after the infusion of endotoxins [8]. On the other hand, the decreased concentration of FFAs due to glucose and insulin infusion was found to be cardio-protective in patients undergoing aortocoronary bypass procedures [9].

Nutritional support plays an essential role in lipid metabolism and one could expect it to have an influence on both the quantitative and qualitative composition of serum FFA in critically ill patients. The FFA concentration in experimental animals being studied increased during the administration of a low carbohydrate, high fat diet, irrespective of their total energy intake [10]. However, neither the total energy intake nor the amounts of macronutrients were found to influence the plasma FFA concentration and the enzymes involved in lipid metabolism in the adipose tissue samples of critical patients who did not survive [11]. Nevertheless, the patients in this “post-mortem” study received a maximum dose of only 250 g of carbohydrates per day.

The question for us was how does nutritional support influence the levels of total plasma FFAs and their qualitative composition in critically ill patients? We also wanted to know what are the effects of TPN on pro–inflammatory and anti–inflammatory FFA. We performed a prospective randomized control study to describe the changes in plasma FFA composition after the administration of two different TPN systems for up to 28 days. The main difference between the TPN regimens that were administered during the course of the study was in the daily dosages of glucose and lipids, which were at the upper and lower limits of the clinical recommendations.

## 2. Materials and Methods

### 2.1. Study Design and Patients

Patients who were acute admissions to the ICU with an expected duration of TPN feeding for more than three days were included in the prospective randomized study. The patients were aged from 18 to 80 years. The ethnicity of the patients enrolled in this study was Caucasian (ethnic Czech). We excluded patients who had received any artificial enteral or parenteral nutrition three days prior to entry into the study. Other exclusion criteria were as follows: pregnancy, body mass index (BMI) less than 18 kg⸱m^−2^, terminal illness, and a baseline serum triglyceride concentration of >3 mmol⸱L^−1^. Patients were randomly divided (randomization using numbered and sealed envelopes) into two groups based on the composition of the TPN admixtures. In the first group, a non-protein energy needs consisted of glucose (group G) and, in the second group, non-protein energy was infused as a fixed mixture of glucose and a lipid emulsion (group L).

The study protocol and the informed consent were approved by the ethical committee of the University Hospital of Charles University in Prague, in the Czech Republic (the official trial number of the study is 1411S22P). Upon everything being explained to them, the patients signed informed consent forms prior to their participation in this study. The primary goal of this study was to determine the influence of TPN on the differences in the total plasma FFA as well as on the individual fatty acids (FA) profiles. The secondary objective of the study was to compare the differences in selected antioxidant (α–Tocopherol) and adipose tissue hormones, along with a detailed glycaemic profile and checking for signs of EFAD in the patients being studied.

### 2.2. Composition of the TPN Admixtures

The nutritional regimes of both group G and L contained an identical amount of energy and amino acids. The planned protein intake was 1.8 g∙kg^−1^ of ideal body weight (IBW) per day (Neonutrin 15%, Fresenius Kabi, Prague, Czech Republic) and the energy intake at 30 kcal g∙kg^−1^ of IBW per day. The only difference was the amount of lipid emulsion and glucose: group G received glucose at a dose of 5.8 g∙kg^−1^ IBW∙day^−1^ (≈400 g per 70 kg BW) without any lipid emulsion. Group L received glucose at a dose of 2.9 g∙kg^−1^ of IBW∙day^−1^ (≈200 g per 70 kg BW) with lipid emulsion at a dose of 1.2 g∙kg^−1^ of IBW∙day^−1^ (SMOF-lipid 20%, Fresenius Kabi, Uppsala, Sweden). The IBW was calculated according to the Devine formula, which is often used in the ICU setting and is based on the height of the patient [12]. The treating physicians, together with members of the nutritional support team, tailored the TPN treatment according to the needs of the patient. The ICU team, in cooperation with the nutritional support team, also determined the timing of the initiation and discontinuation of the TPN regimen and the commencement of enteral feeds. The study investigators themselves were not involved in making these decisions. The TPN admixture was continuously infused during a 24 hour period and the timing of its administration was carefully controlled. The plasma glucose level was maintained at concentrations between 8 and 10 mmol∙L^−1^ and, if necessary, insulin was infused according to our ICU protocols. Electrolytes, vitamins, and trace elements were added in the hospital pharmacy to “all-in-one bags” according to the patient’s needs. The pharmacist also checked the compatibility of the TPN admixtures.

### 2.3. Blood Collection and Sample Analysis

Baseline blood samples were drawn and the results were analyzed 6 h before the start of TPN (day 0). Subsequent samples were taken in the mornings on days 1, 3, 6, 9, 14, and 28. If TPN was ceased or if any form of enteral feeding (enteral nutrition, sipping, and so on) was scheduled, blood samples were taken on the last day of TPN. Blood samples were no longer collected once the patient was discharged from the ICU.

First, 5 mL blood samples were collected into tubes with potassium ethylenediaminetetraacetic acid (1.8 mg∙mL^−1^; Vacutainer, BD Diagnostics, NJ, USA). The samples were transported to the laboratory on ice and then immediately centrifuged at 4 °C at a relative centrifugal force of 2500 rpm for 10 min. The plasma was separated and stored at a temperature of −20 °C. The use of heparin was minimized, or the application was performed at intervals to minimize its effect on lipoprotein lipase. The plasma concentrations of total FFAs were analyzed using an FFA-HR kit (Wako chemicals GmbH, Neuss, Germany) with a Shimadzu UV-1700 spectrophotometer (produced in Kyoto, Japan). The accuracy of this method ranges from 3.3% to 5.1% and the precision is 4.9%. The normal serum range of total FFAs is between 0.1 and 0.45 mmol∙L^−1^ in females and between 0.1 and 0.6 mmol∙L^−1^ in males. FFA values above these ranges were considered to be abnormally high. Individual FAs were analyzed using the gas chromatography technique with a flame ionization detector (Dani Master, Dani Instruments S.p.A., Milan, Italy). Following its extraction from plasma by the usage of toluen–acetylchloride and its derivatization using methanol, chromatographic separation was performed in a Rtx-2330 column (60 m, 0.25 mm ID, 0.2 um df; Restek, Bellefonte, PA, USA). The precision of the gas chromatography method was from 3% to 10%, depending on the low and high range and on the type of fatty acid. The individual FA composition was expressed as a weight percentage of the total FFAs (presented as %). The levels of markers affecting insulin resistance (resistin, leptin), plasminogen activator inhibitor 1 (PAI-1), and insulin levels were measured by biochip array technology using chemiluminescent sandwich immunoassays (Metabolic Syndrome Array I, Randox, Crumlin, UK) that were applied to the evidence investigator analyzer [13]. For quality control in the monitoring of the accuracy and precision control assays (Metabolic Syndrome Multianalyte Controls), EV3757 was used. Assay precision was determined from 4% to 11% depending on the type of hormone. The levels of α–tocopherol were analyzed using a prominence HPLC system (Schimadzu, Kyoto, Japan), as previously published by Solichova et al. [14]. Additional 5 mL blood samples were collected into tubes with silica clot activator (Vacutainer, BD Diagnostics, NJ, USA) and used for further biochemical analysis ( for example albumin). These analytes were determined on a Cobas 8000 analyzer (Hoffmann-La Roche Ltd., Manheim, Germany).

### 2.4. Statistical Analysis

We used the nonparametric analysis of longitudinal data of the F1-LD-F1 design [15] on data generated during the course of the study, and we applied Bonferroni–Holm correction for multiple testing [16]. The analysis was performed in R 3.5.3 2019 [17]. For the purpose of group comparison, the baseline data of the 48 patients who were enrolled in the study were analyzed using the unpaired t-test. Significance was defined by a 5% level and the data were presented as means plus/minus standard deviations (SDs), unless otherwise specified.

## 3. Results

### 3.1. Group Characteristics

A total of 48 patients (37 males and 11 females) with a mean age of 65 years were participants in the study (Appendix A). The overview of the main diagnosis of our cohort at the beginning of study is depicted in Table 1. In both study groups, there were no observable differences in their main characteristics prior to them receiving nutritional support (Table 2). The study population consisted of severely ill patients with a high acute physiology and chronic health evaluation (APACHE) II score in addition to having a high risk of malnutrition according to the nutrition risk score (NRS)-2002 scoring system.

### 3.2. Macronutrient Intake and Plasma Glucose Control

TPN on average was started 2–3 days after admission to the ICU. Energy and amino acid intake did not significantly differ between the groups and the predetermined nutritional goals were met for all of the patients (Table 3). Group G received 5.8 g of glucose∙kg^−1^ of IBW∙day^−1^ without receiving any lipids, whereas group L received 2.9 g of glucose∙kg^−1^ of IBW∙day^−1^ and 1.2 g of lipids∙kg^−1^ of IBW∙day^−1^. In spite of the fact that the glucose and lipid intakes significantly differed between the groups, we did not observe any statistically significant differences in the plasma glucose levels (8.9 ± 1.4 mmol∙l^−1^ vs. 8.0 ± 1.3 mmol∙l^−1^) among the patients. Moreover, the rate of insulin administration was also higher from a negligible to a moderate level in group G, in comparison with group L (68 ± 57 mIU∙day^−1^ vs. 43 ± 36 mIU∙day^−1^). There were two incidents where mild hypoglycaemic states were observed in group G versus none in group L.

### 3.3. Plasma Concentrations of Total Free Fatty Acids

The baseline levels of total plasma FFAs were not significantly different between the study groups and changed rapidly over time (Figure 1). Fourteen patients (29.2%) had higher levels of FFAs (median 0.796 mmol∙L-1) before the start of TPN. Plasma FFA levels changed rapidly after the onset of TPN and during the study period (Figure 1). The significant decrease (*p* < 0.001) in the total FFA concentration occurred in both groups three days after the onset of TPN, but this decrease was significantly higher (*p* < 0.001) in the group that received glucose as the only source of non-protein energy. Over time, the FFA levels further decreased and the lowest serum levels of total FFAs was found on day 28 in both groups (0.12 ± 0.07 vs. 0.27 ± 0.07; Figure 1).

The characteristic and composition of individual FA in plasma were assessed and significant differences were observed in the plasma levels of some individual FAs. Glucose-based TPN without lipid emulsions did not have any effect on the ratio between unsaturated and saturated FA in comparison with the lipid group (Table 4). On the other side, the gradual and significant increase in the fraction of the monounsaturated FA and palmitoleic acid (C16:1; Table 4; *p* < 0.001) was observed in group G. Furthermore, we observed lower levels of the relative fraction of linoleic acid (C18:2 ω-6; Table 4; *p* < 0.001) over time in group G, in comparison with group L. The significant increase of docosahexaenoic acid (C22:6 ω–3; Table 4; *p* = 0.003) over time in comparison with the baseline values and between the groups was observed, together with a significantly higher proportion of the sum of omega 3 and omega 6 FFAs in group L.

The mead acid to arachidonic acid ratio is characteristic for EFAD. This ratio was increased on day 28 in group G, but did not meet the diagnostic threshold that is typical for essential fatty acid deficiency (Figure 2A). This result did not reach significance owing to fewer measurements, although the graph looks more convincing. We did not observe any significant differences in the serum profiles of other FAs (Table 4) and plasma triglycerides (Figure 2B).

### 3.4. Antioxidant and Adipose Tissue Hormones

The plasma concentrations of α-tocopherol are shown in Figure 3. We found a significant increase in α–tocopherols in group L with a maximum value on day 28 compared with the baseline and with their counterparts in group G (Figure 3, *p* < 0.001). In order to better understand the glucose homeostasis, we evaluated plasma hormonal levels of resistin, leptin, and PAI-1 in the whole group (Appendix A) and the subset of diabetic patients (Table 5). We observed a non-significant increasing trend of leptin concentrations in group G, when the biggest difference was evident on the sixth day. We observed a high concentration of resistin in both groups (Table 5).

## 4. Discussion

### 4.1. Plasma Total FFA and Individual FA

The physiological reference range for plasma FFA concentrations in our laboratory ranged from 0.100 to 0.600 mmol∙L^−1^. In the ICU patients, Arabi et al. [7] considered a cut-off value to be 0.45 mmol∙L-1 in females and 0.6 mmol∙L^−1^ in males, as the upper limits of normal. In accordance with their cut off, 29.1% of the patients had elevated levels of FFAs at the start of the study, which was almost the same as the Arabi study (32.8%). The concentration of FFA before TPN was, for the most part, closer to the upper limit of the physiological range (Figure 1), which is in accordance with other findings that showed increased FFA turnover during critical illness [18,19]. The significant decrease in total FFAs was observed after the initiation of TPN (Figure 1). As we observed, the higher dosage of glucose in the TPN admixture led to a more marked decrease in plasma FFAs, so it is plausible that glucose dosage plays an important role in the concentration of FFAs. The positive effect of glucose in patients after cardiac surgery was explained by decreased FFA concentrations during glucose infusion.

In our study, the composition of the TPN admixtures influenced not only the total FFA plasma levels, but also the relative levels of some individual FAs (Table 4 and Figure 2A) during the course of critical illness.

The turnover of plasma FFAs is affected by multiple factors in lipid metabolism in both healthy and diseased states. Physiologically, the mobilization of FFAs from adipose tissue (lipolysis) is increased by stress hormones (catecholamines, corticoids) owing to the activation of hormone sensitive lipase with the subsequent release of FFAs into circulation [20,21]. This effect is typical during fasting, when released FAs serve as an important energy source. Fatty acid mobilization also occurs during critical illness, when the increased production of contra-regulatory hormones predominate [22]. On the other side, lipolysis is reduced mainly by the suppressing effects of insulin on hormone sensitive lipase [23,24].

Intravenous administration of nutrients also influences plasma FFA levels. The administration of lipid emulsion led to increased FFA concentrations in healthy volunteers [8]. The clearance of infused FFAs in lipid emulsion is influence by the activation or inhibition of lipoprotein lipase, which releases FAs from plasma lipoproteins. Many factors influence this important enzyme in ICU patients; a good example is heparin application, which releases lipoprotein lipase into the blood with rapid lipolysis occurring for several hours [25]. The released FFAs are subsequently taken up by tissues, especially adipose tissue. In a post-mortem study, higher lipoprotein lipase activity was observed in samples of adipose tissue in non-surviving critically ill patients in comparison with non-critically ill subjects [11]. The lipoprotein lipase activity did not correlate with the total caloric intake of the patients, nevertheless, the effects of the individual energy substrates were not studied [11]. In the peripheral tissues, the released FFAs are either re-esterified with glycerol phosphate to triglycerides (especially in adipose tissue) or oxidized as a source of energy [19]. During esterification, the presence of glucose is essential because glycerol-phosphate is synthetized from this substrate. In this way, glucose can directly affect the capture of FFAs and their esterification to triglycerides in adipose tissue as well as FFA release [26]. Studies with the aim to distinguish whether oxidized FFAs originate from adipose tissue or directly from lipid emulsion (or both) in critically ill patients are limited. Wolfe et al. used stable isotope tracers in critical patients receiving TPN and showed that 70% of the fat component of adipose tissue energy metabolism could be accounted for by the oxidation of plasma FFA, even during the application of exogenous lipid emulsion. His finding may imply that infused lipids are chiefly stored, rather than being directly used as an energy substrate [20,27]. In that study, the exact quantity of oxidized lipids is not clearly recognized, but as glucose oxidation in peripheral tissues decreases, lipolysis takes place at a rather rapid rate [27].

The amount and types of the major nutrients in the TPN admixtures may affect different metabolic pathways. Decreased glucose oxidation and increased FFA oxidation were observed in a study involving septic animals. The authors hypothesized the important role of insulin resistance during the progression of critical illness [19,28]. Lower plasma FFA concentrations in our study (Figure 1) can thus be explained by a decreased activity of hormone sensitive lipase and the inhibition of lipolysis in adipose tissue, together with an improvement in insulin sensitivity. Moreover, insulin-stimulated activity of lipoprotein lipase may enhance the clearance of lipoproteins and FFAs from circulation, and possibly cause even larger decreases in plasma FFAs, owing to higher glucose doses in group G. It was previously shown that the scheme of insulin administration does not affect the concentration of insulin in circulation. An intensive insulin regimen (targeting lower glycaemia) led to lower endogenous insulin secretion and to an increased adiponectin level and an improvement in insulin sensitivity [11,29]. In our study, both groups of patients received TPN together with continuous insulin infusion via a syringe pump. Surprisingly, plasma insulin levels and the rate of insulin administration were not significantly different between both groups (Table 3) despite the fact that there was approximately a 1.5-fold higher insulin dose in group G in comparison with group L. The high interindividual variations in insulin dosage will require more comprehensive investigation in future. Our data demonstrated that glucose and insulin infused together with lipid emulsion reduced the increase in plasma FFAs that was observed in previous studies. Therefore, we suppose that both TPN admixtures in our study can improve insulin sensitivity, with an even better effect in group G. The larger decrease of plasma FFA in critically ill patients who received a glucose-based TPN could be a result of several factors; that is, moderately increased rate of insulin administration, the absence of lipid emulsion in the TPN admixture, and a higher dose of infused glucose. We cannot determine whether the decrease of plasma FFA was the result of the cessation of lipolysis, a change in lipogenesis, or the oxidation of FFA. However, the decreased level of plasma FFA was a direct result of the parenteral nutrition itself, as during the early phase of critical illness, the spontaneous decrease in plasma FFAs is unlikely to be the result of the inflammatory mediators and stress hormones mentioned above.

The storage and subsequent utilization of endogenous lipid stores for energy metabolisms appears to be extremely important in critical disease states that are prolonged, because it protects proteins and glucose for metabolic purposes not related to energy consumption [30]. According to our results, the decrease in FFA concentration was higher with a cumulative dosage of glucose. The difference between the G and L groups was most pronounced in the late state (day 28), whereas the early state was not statistically significant (Figure 1).

It was previously suggested that administration of lipid emulsion to the critically ill can modulate the immune response throughout the alteration of the ω–3/ω–6 ratio. This modification is also affected by the type and quantity of administered FAs. The lipid emulsion that was infused in our study contained 30% soybean oil, 30% medium chain triglycerides, 25% olive oil, and 15% fish oil. This lipid emulsion was designed specifically to modify the immune response during severe inflammatory reactions through the reduction of ω–6 FFAs and an increase in the intake of ω–3 FFAs, with the aim to modify inflammatory reactions and subsequently decrease the incidence of nosocomial infections, improve survival rates, and decrease hospital stays [31,32]. Conversely, Umpierrez et al. did not find any benefits associated with a decreased intake of ω–6 fatty acids owing to the administration of olive oil lipid emulsion in comparison with those receiving soybean oil lipid emulsion in a group of mixed ICU patients [33].

In our study, neither the TPN given to group G nor to group L altered the main ω–6 arachidonic acid (C20:4)/ω–3 eicosapentaenoic acid (C20:5) ratios in circulating FFAs. We found, however, higher proportions of docosahexaenoic acid (C22:6 ω–3) and α–linoleic acid (C18:2 ω–6) in the lipid group (Table 4). The oleic acid concentration was not different between the groups, despite the fact that group L received higher amounts of oleic acid (C18:1 ω–9) and group G did not receive any FFAs (Table 3). A possible explanation for this finding is that there is a large endogenous pool of fatty acids in adipose tissue and these acids can be mobilized during physiological stress. Moreover, oleic acid can be also synthesized in humans. The extended statistical analysis did not find any significant changes in the proportion of saturated or monounsaturated FAs between our groups. There was found to be a significantly higher proportion of the sums of ω–3 and ω–6 fatty acids in group L, and this was mainly because of a higher proportion of linoleic acid (C18:2 ω–6) (Table 4). Experimental studies on rats showed that ω–3 and ω–6 fatty acid supplementation can inhibit de novo lipogenesis [34,35], which is also increased by a fat-free diet [36]. Increased de novo lipogenesis is characterized by a high proportion of circulating palmitic and stearic acids and increased desaturase 9 activity [34]. No such alterations were apparent in our study (Table 4).

The provision of specific types of TPN admixtures for several weeks significantly changed the concentrations of total FFAs in plasma. From a clinical point of view, it has been reported that high concentrations of FFA are present in patients with acute respiratory distress syndrome or sepsis [7,37]. Thus, FFA modulation may be important in positively influencing the inflammatory markers; however, the exact mechanism is unclear. The relative proportions of a large majority of the monitored FFA, even when compared with modern lipid emulsions in the TPN admixtures, were without significant changes. We also did not confirm the previous findings of a modification of the ω–3/ω–6 ratios by lipid emulsion. The heterogenity of the study populations, the types, and the dosage of the nutrients, as well as the treatment duration, are all potential confounders and sources of bias that potentially contributed to the variability of the result of the studies mentioned above. The complex metabolism of polyunsaturated FAs and the unpredictability of which pro-inflammatory and anti-inflammatory end products would result make the generalization of the results more difficult [38]. We hypothesize that the amount and composition of endogenous fat stores appear to be decisive for the composition of the individual FAs pool during the first weeks of a critical disease state.

### 4.2. Adipose Tissue Derived Hormones and Glucose Homeostasis

Adipose tissue is an important endocrine organ that releases hormones and cytokines that affect both the body’s metabolism and immune functions [39]. In this study, we measured the concentrations of leptin and resistin, which are secreted by adipose tissue and play major roles in glucose metabolism; leptin increases, whereas resistin decreases insulin sensitivity [40]. However, there are only a few studies showing the relationship between resistin and leptin in critically ill patients. Increased levels of resistin were associated with poor outcomes in ICU patients and correlated with the presence of inflammatory markers [39,41]. In our study, we found the plasma resistin concentrations in the subset of diabetic patients higher than the physiological range (i.e., <10 ng∙ml^−1^) (Table 5). No statistically significant difference of plasma resistin concentration was apparent during the course of both types of TPN. On the other side, we found a non-significant increase in plasma leptin in group G in comparison with group L, for up to 28 days, the biggest difference was observed on day 6 of TPN administration (Table 5). A similar result was made by McCowen et al., who described an increase in leptin concentration three days after the initiation of TPN [42]. These results may indicate a role of this hormone in insulin resistance in ICU patients.

Despite the big differences in glucose dosage during TPN, there were no significant differences in the insulin infusion rates and the insulin plasma levels between both the G and L groups (Table 3). This may suggest that a majority of the provided glucose is metabolized in non-insulin dependent pathways and that glycaemia and insulin requirements are not closely connected with the actual glucose dosage. According to our data, glucose in TPN does not worsen the insulin resistance of ICU patients during the progression of critical illness for both diabetic as well as non-diabetic patients (Table 3).

### 4.3. Metabolic Alterations

There are several metabolic side effects that are associated with intravenous nutrition. Long periods of a fat-free diet can lead to EFAD. This condition is very rare and more often seen in adults after having undergone complicated gastrointestinal tract surgery [43]. The development of EFAD is usually preceded by biochemical abnormalities, namely alterations in the triene/tetraene ratio (>0.4 for EFAD), the accumulation of mead acid (20:3 ω-9), and decreased concentrations of linoleic acid (C18:2, ω-6) [44]. In our study, we found a gradual decrease in linoleic acid (C18:2, ω-6) in group G and a trend of increased triene/tetraene ratios up to the 28th day of the study (Table 4), suggesting a need for caution when administering long-term TPN without a lipid component. Low doses of lipid emulsions (e.g., 100 g once per week) in the TPN admixture can potentially decrease the incidence of EFAD, as indicated in the guidelines [4].

Hypertriglyceridemia is a relatively frequent complication associated with the use of parenteral nutrition with high amounts of macronutrients. However, several simultaneous critical conditions make the distinction between the changes caused by organ hypoperfusion and the changes due to the side effects of TPN almost impossible to determine [45]. In our study, triglyceride concentrations were in the normal range during the whole period and there were no significant differences between the study groups (Figure 2B).

Parenteral nutrition can be also associated with coagulation disorders, such as fibrinolytic system activation. A correlation between LE and PAI-1 reduction was previously found [46]. In this study, we observed only non-significant trends towards decreased PAI-1 in group L and towards an increase in group G (Table 5). This corresponded with the results of Van der Poll, who did not find a significant effect of TPN on the PAI-1 level in plasma [47].

Low concentrations of vitamin E and other antioxidants have been reported in patients on TPN. Vitamin E protects against the oxidation of polyunsaturated FA, and thus may contribute to the stabilization of cell membranes, and α-tocopherol is the compound with the highest vitamin E activity [48]. In our study, group L received an average of four times the dose of α-tocopherol compared with group G (86 mg vs. 20 mg), because of its inclusion in the lipid emulsion. We observed a gradual and statistically significant increase in α-tocopherol concentration with values ranging between 30 and 40 µmol∙L^−1^ on day 28 in group L (Figure 3). These values were well above the vitamin E deficiency threshold of 11.6 µmol∙L^−1^ [48]. Surprisingly, the α-tocopherol concentration in group G did not change over time, and was above the deficiency threshold until day 28. These results indicate that there was a sufficient supplementation of α-tocopherol in group G, who received vitamin supplementation (Cernevit); however, we cannot exclude vitamin E deficiency in tissues despite normal serum levels, which were previously reported [49].

## 5. Conclusions

The administration of both glucose and lipid plus glucose TPN admixtures resulted in a significant decrease in plasma FFAs. This decrease was more pronounced in lipid-free TPN. Glucose administration did not increase insulin resistance in critically ill patients. We also found changes in the types of individual FAs. After 28 days of lipid-free TPN, the biomarkers of EFAD were apparent. Therefore, we advise to include a lipid emulsion at the latest from three weeks of TPN to prevent EFAD syndrome. Larger prospective, randomized trials with comparable TPN regimens are required to further evaluate the influence of different glucose and lipid doses on the inflammatory markers and clinical outcomes.

## Figures and Tables

**Figure 1 nutrients-12-01373-f001:**
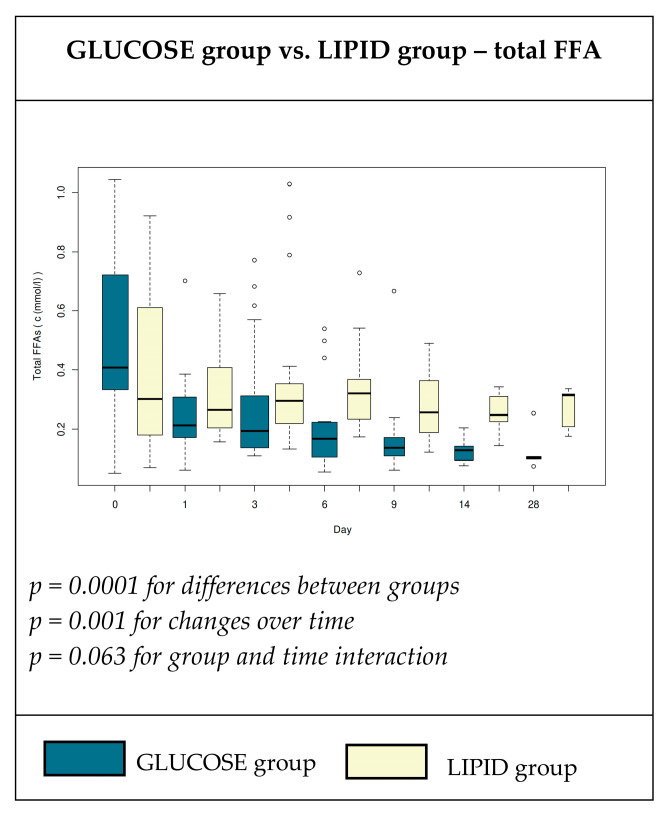
Plasma concentrations (mmol∙L^−1^) of total free fatty acids (FFAs) in patients receiving high glucose; lipid-free (glucose group); or low glucose, high lipid (lipid group) total parenteral nutrition (TPN) admixtures.

**Figure 2 nutrients-12-01373-f002:**
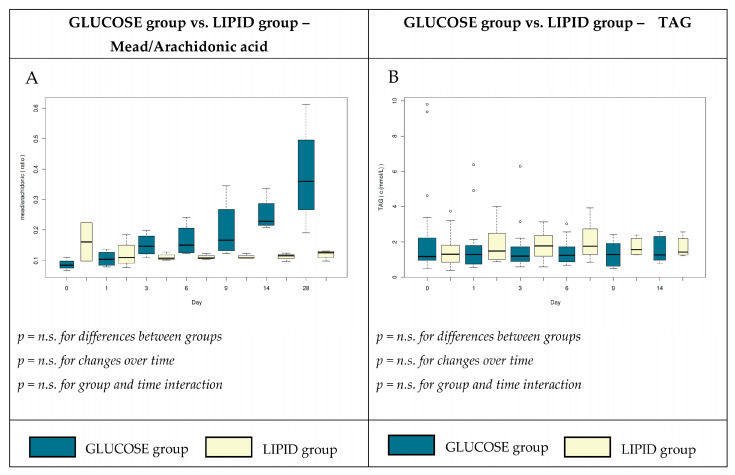
Mead acid to arachidonic acid ratio (**A**) and plasma concentrations (mmol∙L^−1^) of triglycerides (TAG). (**B**) in patients receiving high glucose; lipid-free (glucose group); or low glucose, high lipid (lipid group) TPN admixtures. n.s., non-significant.

**Figure 3 nutrients-12-01373-f003:**
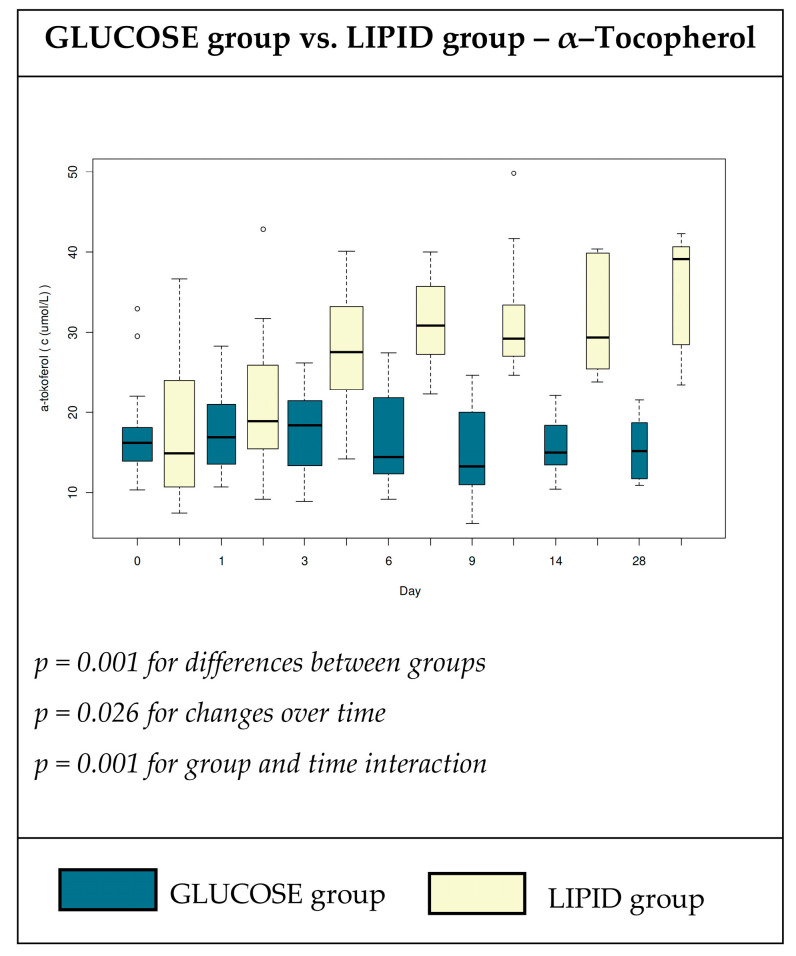
Plasma concentrations (μmol·L^−1^) of a α-tocopherol in patients receiving high glucose; lipid-free (glucose group); or low glucose, high lipid (lipid group) TPN admixtures.

**Table 1 nutrients-12-01373-t001:** Diagnosis of the patients in the study.

Diagnosis	Number of Patients
Acute pancreatitis	22
Gastroinstestinal bleeding	8
Bowel perforation, enteric fistulas	5
Bowel obstruction and ileus	4
Ischemic bowel disease	3
Crohn’s disease	3
Vasculitis with ileus	2
Chemical burns of the oesophagus and stomach	1

**Table 2 nutrients-12-01373-t002:** Group characteristics at day 0 (mean values with standard deviation).

Variable	Group G *n* = 25	Group L *n* = 23	*p*-Value
Age (years)	65 ± 9	65 ± 13	0.609
Male/Female	20/5 (M/F)	17/6	0.528
Weight (kg)	82.4 ± 19.0	82.2 ± 16.0	0.115
BMI (kg∙m^−2^)	28.3 ± 4.9	26.2 ± 3.8	0.105
NRS-2002	4.2 ± 0.7	4.1 ± 0.9	0.404
APACHE II score	20.8 ± 5.9	18.4 ± 4.0	0.144
Diabetic patients, # (% of total)	9 (36)	10 (43)	0.365
Serum albumin (g∙L^−1^)	29.8 ± 8.2	29.3 ± 7.4	0.185
Serum transthyretin (mmol∙L^−1^)	0.13 ± 0.06	0.16 ± 0.06	0.185
Serum triacylglycerols (mmol∙L^−1^)	1.5 ± 0.9	1.6 ± 0.9	0.704
Serum total cholesterol (mmol∙L^−1^)	3.1 ± 1.3	3.1 ± 1.2	0.378
Serum HDL cholesterol (mmol∙L^−1^)	0.7 ± 0.5	0.8 ± 0.5	0.905
Leukocyte count (cells∙109∙L^−1^)	12.6 ± 6.9	13.4 ± 7.0	0.840
C-reactive protein [mg∙L-^1^)	174.0 ± 163.3	91.0 ± 79.8	0.311
Serum ALT(µkat∙ L^−1^)	1.9 ± 2.3	1.0 ± 1.2	0.285
Serum AST (µkat∙ L^−1^)	3.1 ± 4.7	1.1 ± 1.9	0.111
Serum GMT(µkat∙L^−1^)	4.7 ± 5.3	1.3 ± 2.6	0.178
Serum ALP (µkat∙L^−1^)	2.0 ± 1.4	1.8 ± 1.4	0.401

Group G, glucose-based total parenteral nutrition; Group L, lipid-based total parenteral nutrition; BMI, body mass index; NRS, nutrition risk score; APACHE, acute physiology and chronic health evaluation; HDL, high density lipoprotein; ALT, Alanin–aminostransferase; AST, Aspartate–aminotransferase; GMT, Gamma–glutamyltransferase; ALP, Alanine-aminotransferase.

**Table 3 nutrients-12-01373-t003:** Nutritional intake and glycaemic control throughout the course of total parenteral nutrition (mean values with standard deviation).

Variable	Group G *n* = 25	Group L *n* = 23	*p*-Value
Duration of TPN (days)	14 ± 11	11 ± 8	0.704
Starting TPN from the time of admission (days)	2.3 ± 0.8	2.1 ± 0.7	0.680
Energy (kcal∙kg^−1^ IBW∙day^−1^)	30.5 ± 1.8	30.4 ± 1.3	0.764
Non-protein energy (kcal∙kg^−1^ IBW∙day^−1^)	23.2 ± 1.6	23.1 ± 1.0	0.788
Amino acids (g∙kg^−1^ IBW∙day^−1^)	1.8 ± 0.1	1.8 ± 0.1	0.775
Glucose (g∙kg^−1^ IBW∙day^−1^)	5.8 ± 0.4	2.9 ± 0.2	<0.001
Lipids (g∙kg^−1^ IBW∙day^−1^)	0	1.2 ± 0.1	<0.001
Oleic acid received (g∙day^−1^)	0	21.7 ± 2.4	<0.001
Linoleic acid received (g∙day^−1^)	0	14.6 ± 1.6	<0.001
Blood glucose (mmol∙L^−1^)	8.9 ± 1.4	8.0 ± 1.3	0.115
Blood glucose in diabetic patients (mmol∙L^−1^)	9.3 ± 1.1	9.1 ± 1.0	0.682
Presence of mild hypoglycaemia (3–3.8 mmol∙L^−1^; cases)	2	0	0.345
Dose of insulin (mIU∙day^−1^)	68 ± 57	43 ± 36	0.205
Dose of insulin in diabetic patients (mIU∙day^−1^)	82 ± 69	71 ± 43	0.418
Administration of insulin (% of patients)	68	52	0.314
Serum insulin levels (μIU∙mL^−1^)	40 ± 59	25 ± 44	0.215
Serum insulin levels in diabetic patients (μIU∙mL^−1^)	43 ± 71	36 ± 32	0.575

Group G, glucose-based total parenteral nutrition; Group L, lipid-based total parenteral nutrition; TPN, total parenteral nutrition.

**Table 4 nutrients-12-01373-t004:** Individual fatty acids (weight percentage of total fatty acids, %).

Variable	Group	Day 0	Day 1	Day 3	Day 6	Day 9	Day 14	Day 28	*p*-Value
C12:0	GG	0.22 ± 0.23	0.13 ± 0.10	0.15 ± 0.11	0.14 ± 0.08	0.14 ± 0.06	0.14 ± 0.10	0.10 ± 0.02	n.s.
	LL	0.12 ± 0.05	0.11 ± 0.05	0.10 ± 0.04	0.11 ± 0.07	0.11 ± 0.10	0.11 ± 0.04	0.09 ± 0.03	n.s.
C14:0	GG	1.69 ± 0.78	1.38 ± 0.86	1.69 ± 0.71	1.61 ± 0.60	1.79 ± 0.81	2.06 ± 1.35	1.57 ± 0.62	n.s.
	LL	1.21 ± 0.69	1.13 ± 0.56	1.16 ± 0.65	1.17 ± 0.71	1.05 ± 0.72	1.35 ± 0.93	0.97 ± 0.47	n.s.
C16:0	GG	32.82 ± 4.87	30.96 ± 3.78	32.07 ± 4.81	31.09 ± 3.85	31.59 ± 4.87	31.85 ± 5.74	30.28 ± 4.17	n.s.
	LL	30.20 ± 4.62	29.39 ± 4.05	28.45 ± 4.55	27.78 ± 4.88	26.61 ± 3.86	27.08 ± 4.14	25.81 ± 2.20	n.s.
C16:1	GG	3.91 ± 2.14	4.16 ± 2.09	4.73 ± 2.14	5.69 ± 2.10	5.97 ± 2.04	7.61 ± 2.28	8.23 ± 3.21	*
	LL	3.08 ± 1.53	3.05 ± 1.17	2.80 ± 1.00	2.43 ± 0.59	2.37 ± 0.69	2.36 ± 0.61	1.76 ± 0.57	*
C18:0	GG	7.25 ± 1.46	8.03 ± 1.40	7.42 ± 1.32	7.33 ± 1.30	7.37 ± 1.70	6.73 ± 2.07	7.67 ± 1.65	n.s.
	LL	7.62 ± 1.65	7.94 ± 1.39	8.01 ± 1.37	7.65 ± 1.27	7.79 ± 0.93	8.22 ± 1.42	8.65 ± 1.24	n.s.
C18:1	GG	29.31 ± 5.76	32.39 ± 6.08	32.00 ± 5.70	34.22 ± 4.89	33.83 ± 5.79	34.29 ± 4.50	35.29 ± 1.89	n.s.
	LL	29.96 ± 4.80	32.65 ± 4.86	31.12 ± 4.26	30.89 ± 3.34	30.53 ± 4.47	29.62 ± 2.80	29.38 ± 3.40	n.s.
C18:2 n6	GG	17.22 ± 5.64	14.28 ± 7.78	13.36 ± 6.62	10.96 ± 4.90	10.54 ± 5.59	9.18 ± 4.13	8.29 ± 3.35	**
	LL	18.71 ± 7.59	16.75 ± 9.89	19.12 ± 6.73	21.45 ± 1.95	21.77 ± 1.92	21.91 ± 1.63	22.99 ± 2.88	**
C18:3 n6	GG	0.34 ± 0.21	0.41 ± 0.30	0.74 ± 0.21	0.43 ± 0.23	0.48 ± 0.33	0.48 ± 0.36	0.43 ± 0.27	n.s.
	LL	0.30 ± 0.17	0.30 ± 0.20	0.34 ± 0.19	0.34 ± 0.20	0.33 ± 0.15	0.35 ± 0.18	0.31 ± 0.20	n.s.
C18:3 n3	GG	0.30 ± 0.24	0.49 ± 0.30	0.71 ± 0.40	0.86 ± 0.47	0.71 ± 0.36	0.72 ± 0.33	0.64 ± 0.26	n.s.
	LL	0.42 ± 0.21	0.46 ± 0.23	0.46 ± 0.29	0.49 ± 0.33	0.45 ± 0.24	0.50 ± 0.18	0.59 ± 0.43	n.s.
C20:3 n6	GG	0.79 ± 0.28	1.06 ± 0.39	1.28 ± 0.53	1.48 ± 0.58	1.51 ± 0.64	1.32 ± 0.60	1.54 ± 0.65	n.s.
	LL	1.10 ± 0.64	1.04 ± 0.46	1.01 ± 0.33	0.81 ± 0.28	0.86 ± 0.18	0.86 ± 0.35	0.82 ± 0.24	n.s.
C20:4 n6	GG	4.59 ± 1.87	4.90 ± 1.97	4.62 ± 1.88	4.50 ± 1.81	4.45 ± 1.94	4.02 ± 2.28	4.33 ± 2.04	n.s.
	LL	5.22 ± 1.95	4.94 ± 1.51	4.82 ± 1.72	4.19 ± 1.50	4.74 ± 1.42	4.29 ± 1.47	4.31 ± 1.35	n.s.
C20:5 n3	GG	0.69 ± 0.43	0.79 ± 0.48	0.73 ± 0.47	0.89 ± 0.46	0.81 ± 0.38	0.90 ± 0.29	0.89 ± 0.43	n.s.
	LL	0.88 ± 0.46	0.73 ± 0.32	0.89 ± 0.37	0.98 ± 0.40	0.99 ± 0.34	1.05 ± 0.52	1.02 ± 0.65	n.s.
C22:6 n3	GG	0.89 ± 0.48	1.00 ± 0.52	0.79 ± 0.46	0.79 ± 0.47	0.80 ± 0.48	0.69 ± 0.42	0.74 ± 0.53	***
	LL	1.19 ± 0.62	1.50 ± 0.87	1.73 ± 0.88	1.73 ± 1.00	2.40 ± 1.12	2.31 ± 1.23	3.29 ± 1.52	***
AA to EPA	GG	8.88 ± 5.61	8.29 ± 5.59	8.41 ± 5.02	6.17 ± 3.78	6.75 ± 4.84	4.71 ± 2.59	5.68 ± 2.78	n.s.
	LL	7.64 ± 4.41	11.2 ± 17.47	6.37 ± 4.38	4.93 ± 3.03	5.67 ± 3.54	5.26 ± 3.64	6.44 ± 5.29	n.s.
n3 + n6	GG	24.52 ± 6.99	22.63 ± 7.31	21.55 ± 6.53	19.59 ± 5.03	18.94 ± 6.17	16.96 ± 5.62	16.64 ± 4.54	****
	LL	27.65 ± 8.06	25.57 ± 8.71	28.17 ± 6.67	29.79 ± 4.16	31.38 ± 3.73	31.06 ± 3.96	33.17 ± 5.04	****

GG = group G, glucose-based total parenteral nutrition; LG = group L, lipid-based total; mean values ± standard deviation; n.s., non-significant, AA to EPA = C20:4 n–6/C20:5 n–3; * *p* < 0.001 for differences between groups, *p* = n.s. for changes over time, *p* < 0.001 for group and time interaction; ** *p* < 0.001 for differences between groups, *p* = n.s. for changes over time, *p* < 0.001 for group and time interaction; *** *p* = 0.003 for differences between groups, *p* = n.s. for changes over time, *p* = n.s. for group and time interaction; **** *p* = 0.001 for differences between groups, *p* = n.s. for changes over time, *p* = 0.001 for group and time interaction.

**Table 5 nutrients-12-01373-t005:** Plasma concentrations of selected hormones of adipose tissue in the subset of diabetic patients (mean values with standard deviations).

Variable	Group	Day 0	Day 1	Day 3	Day 6	Day 9	Day 14	Day 28	*p*-Value
Resistin (ng·mL^−1^)	GG	27.76 ± 6.86	26.28 ± 8.51	27.34 ± 7.64	24.36 ± 9.35	25.79 ± 7.11	27.97 ± 4.69	25.3 ± 9.21	n.s.
	LG	25.83 ± 9.97	25.29 ± 9.5	23.88 ± 9.57	27.45 ± 6.46	24.46 ± 9.69	26.94 ± 9.99	24.55 ± 6.22	n.s.
Leptin (pg·mL^−1^)	GG	4.86 ± 3.16	24.34 ± 11.9	20.49 ± 13.9	17.17 ± 11.47	21.48 ± 12.72	21.99 ± 9.73	16.69 ± 7.12	n.s.
	LG	4.34 ± 4.11	4.45 ± 5.05	4.37 ± 3.52	2.98 ± 2.52	5.77 ± 6.69	4.19 ± 5.08	5.17 ± 6.20	n.s.
PAI-1 (ng·mL^−1^)	GG	73.77 ± 38.59	90.66 ± 61.61	72.25 ± 40.21	66.33 ± 26.25	104.65 ± 82.71	85.25 ± 79.10	78.11 ± 57.23	n.s.
	LG	50.39 ± 28.49	41.92 ± 25.34	46.21 ± 30.11	51.36 ± 46.31	68.65 ± 54.59	55.46 ± 33.67	51.18 ± 29.40	n.s.

GG = group G, glucose-based total parenteral nutrition; LG = group L, lipid-based total parenteral; PAI-1, plasminogen activator inhibitor-1; n.s., non-significant.

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
