# Peer review of "The Impact of Glucose-Based or Lipid-Based Total Parenteral Nutrition on the Free Fatty Acids Profile in Critically Ill Patients"

_nutrients, 2020, doi:10.3390/nu12051373_

Round 1

Reviewer 1 Report

In the current manuscript, the authors have performed a controlled randomized clinical trial study to investigate the influence of nutritional support on the levels of total plasma free fatty acid (FFA) in critically ill patients. A total of 48 patients completed the study; these patients were divided into two study groups i.e. Group G (Non-protein energy constituted of glucose) and Group L (Non-protein energy constituted of glucose and lipid emulsion). Overall, the manuscript is well-written and clinical trial well designed. Kindly find my comments below:

Major Comments:

  1. Authors kindly provide references for the analytical method employed to estimate the FFA and other biochemical levels investigated in this manuscript. If it is an in-house method, kindly add the precision and accuracy data for the assay to the manuscript.
  2. Data from table 2 suggests that less than 50% in both the groups G & L were diabetic patients at the initiation of the trial. It would be best to consider only these subsets of patients for evaluating the plasma hormonal levels in order to better understand the glucose homeostasis. Kindly provide a separate table for plasma hormonal concentrations for diabetic patients.

Minor Comments:

  1. In Table 2, several biochemical parameters have been investigated using serum. Kindly add a sentence or two on serum sample collection and processing.
  2. Authors kindly follow a uniform referencing style throughout the manuscript. Doi is missing for references 14, 16, 17, 18, 25, 39 and 41. Also, abbreviated journal names are used in some references while some references have the full journal name.
  3. What was the ethnicity of the patients enrolled in this trial?

Reviewer 2 Report

Comments to the Author
Dear Authors, thank you for the opportunity to review your manuscript. I hope that the following comments are helpful to you.
This study is important and enriches the knowledge of artificial nutrition.
Overall, the manuscript is well written and the data are comprehensive.
Recommend reviewing the use of the Total Parenteral Nutrition which is treatment procedure, and no parenteral nutrition admixture. Sometimes I not sure the authors distinguish these differences.
line 108. IBW vas change into was
Paragraph 2.3 The authors used abbreviation for liter l and L, please make it concise
In discussion should be more described the role of fatty acid on inflammatory and how does it result in ICU patient.

Reviewer 3 Report

Can the authors please elucidate that for all the free-fatty acid patterns they tracked, how did they control that the observations were not post-heparin lipoprotein lipase activity?  Patients in the ICU have central lines flushed with heparin, they receive Intravenous Heparin, receive Heparin indirectly through the exta-corporeal circuits of dialysis machines (IHD, SLED, CVVHD etc).   How have the authors safe-guarded against the randomly time administrations of heparin which would provoked a volley of post-heparin lipoprotein effect?

I note that non of the citations listed broach the topic of heparin lipo-protein activity and to that end, I am wondering if the fundamental study design controlled for this confounder.

Round 2

Reviewer 3 Report

Some progress made in addressing the concerns expressed.